# FIDE: Frequency-Inflated Conditional Diffusion Model for Extreme-Aware Time Series Generation

**Asadullah Hill Galib, Pang-Ning Tan, and Lifeng Luo**
Michigan State University
Emails: {galibasa, ptan, lluo}@msu.edu

## Abstract

Time series generation is a crucial aspect of data analysis, playing a pivotal role in learning the temporal patterns and their underlying dynamics across diverse fields. Conventional time series generation methods often struggle to capture extreme values adequately, diminishing their value in critical applications such as scenario planning and risk management for healthcare, finance, climate change adaptation, and beyond. In this paper, we introduce a conditional diffusion model called `FIDE` to address the challenge of preserving the distribution of extreme values in generative modeling for time series. `FIDE` employs a novel high-frequency inflation strategy in the frequency domain, preventing premature fade-out of the extreme values. It also extends the traditional diffusion-based model, enabling the generation of samples conditioned on the block maxima, thereby enhancing the model's capacity to capture extreme events. Additionally, the `FIDE` framework incorporates the Generalized Extreme Value (GEV) distribution within its generative modeling framework, ensuring fidelity to both block maxima and overall data distribution. Experimental results on real-world and synthetic data showcase the efficacy of `FIDE` over baseline methods, highlighting its potential in advancing Generative AI for time series analysis, specifically in accurately modeling extreme events.

## 1 Introduction

Generative models [18, 10, 13] have revolutionized the AI landscape, demonstrating their broad applicability across diverse domains, including computer vision and natural language processing. Such models are designed to learn the underlying data distribution and exhibit resilience to overfitting while promoting automatic feature extraction. Diffusion-based models [12, 19], in particular, have emerged as a popular generative AI method due to their capability to generate realistic, high-quality data. This paper examines the application of diffusion-based models for time series generation. In particular, we investigate the following issue: *How well do existing diffusion models preserve the fidelity of extreme values (i.e., tail distribution) of the original time series?*

The modeling of extreme values in time series is essential for informed decision-making across diverse applications, including weather forecasting, earthquake prediction, and disease outbreak detection. Effective generative modeling of these extremes is important as it aids in learning the underlying data distribution, facilitating data augmentation, and improving uncertainty estimation, all of which are crucial for developing robust risk management strategies and enhancing disaster preparedness measures. While there has been growing research on applying diffusion models for time series [20, 2], their ability to preserve the distribution of extreme values remains largely underexplored. In this study, we examine how effectively diffusion models preserve extreme values in the form of block maxima [4], defined as the peak value within a specified time window.

To illustrate the difficulty of modeling the distribution of block maxima, Figure 1 shows the result of applying the Denoising Diffusion Probabilistic Model (DDPM) [12] to a synthetic AR(1) dataset.

38th Conference on Neural Information Processing Systems (NeurIPS 2024).

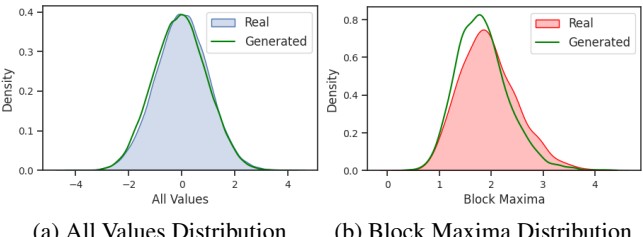

(a) All Values Distribution      (b) Block Maxima Distribution

Figure 1: Comparing the distributions of all values and block maxima values for real and generated samples using DDPM [12] when applied to the synthetic AR(1) dataset.

While DDPM shows proficiency in generating samples that closely align with the overall data distribution (left diagram), it struggles to preserve the distribution of block maxima values (right diagram) when the generated time series is partitioned into disjoint time windows.

In this paper, we identify the key shortcomings of existing diffusion models that hamper their ability to accurately model block maxima values. We then present a novel framework to overcome this limitation. Our key observation is that unusually large block maxima values, often linked to abrupt temporal changes, are strongly associated with high-frequency components of the time series. As the diffusion-based generative model gradually introduces noise with a linearly increasing variance schedule, it slowly diminishes the long-term trends (low-frequency components) of the time series while quickly attenuating the high-frequency components. These high-frequency components are crucial for reproducing extreme block maxima values. This limitation hampers the accurate representation of the block maxima, necessitating the development of new techniques.

To address this challenge, we propose an end-to-end diffusion model framework termed FIDE. First, to mitigate the rapid dissipation of high-frequency components in the diffusion model, we introduce a novel high-frequency inflation strategy within the frequency domain. This strategic augmentation ensures the sustained emphasis on block maxima, preventing their premature fade-out. We further employ a conditional diffusion-based generative modeling approach to guide the time series generation by conditioning on their block maxima. To enhance the preservation of the block maxima distribution while learning the overall data distribution, we extend the conventional framework with a regularization term in the loss function based on the negative log-likelihood of the Generalized Extreme Value (GEV) distribution. Using these strategies, we empirically show that our approach effectively addresses the challenges of learning the overall data distribution while simultaneously preserving the block maxima distribution.

## 2 Preliminaries

Consider a time series dataset $\mathcal{D} = \{\mathbf{x}_{m,0}\}_{m=1}^{M}$ comprising of $M$ samples, where each sample $\mathbf{x}_{m,0} = (x_{m,0}^1, x_{m,0}^2, \cdots, x_{m,0}^T)$ is a univariate time series of finite length $T$. Let $\mathbf{f}_{m,0} \in \mathbb{R}^T$ be the Fourier coefficients, whose $k$-th frequency component is obtained by applying the following discrete Fourier transform on $\mathbf{x}_{m,0}$:

$$f_{m,0}^k = \sum_{t=1}^{T} x_{m,0}^t \, e^{-i2\pi tk/T} = \sum_{t=1}^{T} \left[ x_{m,0}^t \cos\left(\frac{2\pi tk}{T}\right) - i \cdot x_{m,0}^t \sin\left(\frac{2\pi tk}{T}\right) \right] \tag{1}$$

The time series can be recovered from its Fourier coefficients using the following inverse discrete Fourier transform:

$$x_{m,0}^t = \frac{1}{T} \sum_{k=1}^{T} f_{m,0}^k \, e^{i2\pi tk/T} = \frac{1}{T} \sum_{k=1}^{T} \left( f_{m,0}^k \cos\left(\frac{2\pi tk}{T}\right) + i \cdot f_{m,0}^k \sin\left(\frac{2\pi tk}{T}\right) \right) \tag{2}$$

For brevity, we will drop the sample subscript $m$ when it is clear from the context. Let, $\omega_k = \frac{2\pi k}{T}$ be the $k$-th frequency in Fourier transform.

Given a sample $\mathbf{x}_0$, let $y_0$ be its corresponding block maxima value, where $y_0 = \max_{\tau \in \{1, \cdots, T\}} x_0^\tau$. The distribution of the block maxima values is governed by the Generalized Extreme Value (GEV)

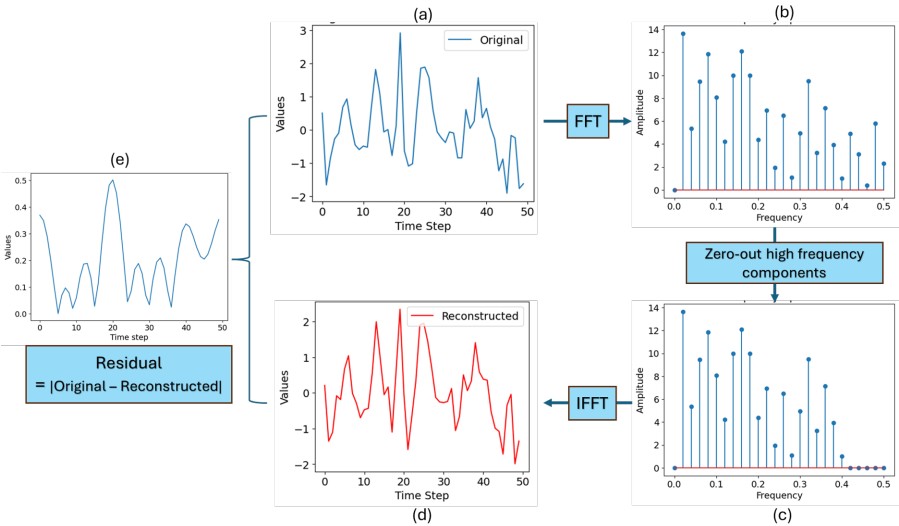

Figure 2: Removal of high-frequency components from daily temperature time series significantly alters the magnitude of its block maxima value (at time step 20), as evidenced by its high residual.

distribution, whose cumulative distribution function is given as follows [4]:

$$G(y) = \exp\left\{ -\left[1 + \xi\left(\frac{y-\mu}{\sigma}\right)\right]^{-1/\xi} \right\} \tag{3}$$

where $\mu$ (location), $\sigma$ (scale), and $\xi$ (shape) are the distribution parameters. Given $M$ independent block maxima values, denoted as $\{y_{1,0}, y_{2,0}, \cdots, y_{M,0}\}$, with the cumulative distribution function given by Equation (3), the distribution parameters can be estimated using the maximum likelihood approach by minimizing the following negative log-likelihood function:

$$-\log\mathcal{L}_{\text{GEV}}(\mu,\sigma,\xi) = M\log\sigma + \left(\frac{1}{\xi}+1\right)\sum_{i=1}^{M}\log\left[1 + \xi\frac{y_{i,0}-\mu}{\sigma}\right] + \sum_{i=1}^{M}\left(1 + \xi\frac{y_{i,0}-\mu}{\sigma}\right)^{-1/\xi} \tag{4}$$

## 3 On the Rapid Dissipation of Block Maxima in Diffusion Models

While diffusion models have demonstrated remarkable capabilities in learning complex data distributions, a significant challenge arises in accurately capturing the distribution of block maxima values, as evidenced by Figure 1. Addressing this shortcoming is crucial for enhancing the performance and applicability of these models across various domains. In this section, we delve into the root cause of this phenomenon and present insightful observations that shed light on the underlying issue.

Our first key observation reveals a ***connection between block maxima with abrupt changes and the high-frequency components*** of many real-world time series. Block maxima, often characterized by their rarity and abrupt temporal changes, are intrinsically linked to the high-frequency components of the data. This relationship is observed in many real-world datasets, where the block maxima values do not typically evolve smoothly but rather emerge through large deviations from their adjacent values.

To illustrate this, consider the real-world temperature time series depicted in Figure 2. In this plot, we first transform the time series into its Fourier domain, obtaining its frequency components, and selectively zeroing out its top-5 highest frequency components. We then reconstruct the time series via its inverse Fourier transform and compute the difference between the original and reconstructed time series. The recovered signal exhibits a notable distortion around the block maxima value, as evidenced by the larger residual at time step 20, where the block maxima value occurs. This suggests that the removal of high-frequency components of a time series has a significant impact on the

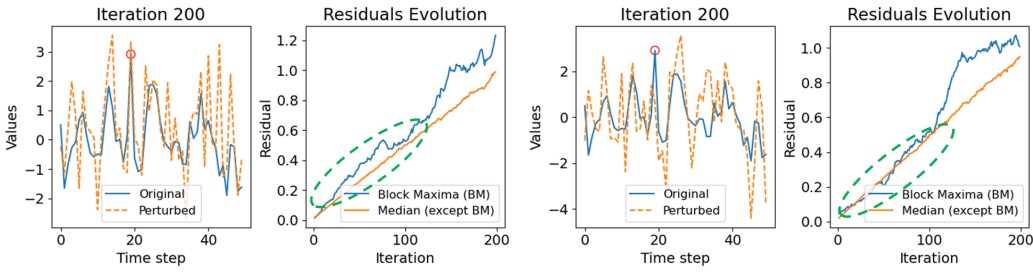

(a) Attenuation of Block Maxima by DDPM  (b) Effect of High Frequency Inflation

Figure 3: A comparison of the effects of noise addition by existing DDPM versus high-frequency inflation on the block maxima of generated samples.

accurate representation of block maxima values. A more detailed analysis supporting this argument is given in Appendix B.

Our second key observation unveils a concerning behavior of diffusion models: ***the addition of noise diminishes high frequency components, i.e., block maxima, at a faster rate*** compared to other values in the signal. As diffusion-based generative models gradually introduce noise characterized by a linearly increasing variance scheduler, they inadvertently attenuate the signals associated with high-frequency components. These components, as established in our first observation, are crucial for accurately reproducing block maxima. Concurrently, the models effectively capture the long-term trends and low-frequency components, which are conducive to learning the overall data distribution. However, the high frequency components dissipate more rapidly, hindering the model's ability to adeptly learn the distribution of the block maxima values.

Figure 3a illustrates this phenomenon. By tracking the evolution of residuals, or the differences between the original and perturbed time series generated by DDPM, we observe a discernible pattern: block maxima dissipate at a faster rate compared to other values, as evidenced by the higher residuals associated with these extreme points. Notably, in the early iterations highlighted by the green circle, the substantially higher residual suggests that the block maxima signal is rapidly transformed into noise, outpacing the dissipation rate of other values. This behavior poses a formidable challenge for diffusion models in effectively capturing the distributions of the block maxima values.

To substantiate our observations, the theorem below offers a rigorous justification for the rapid dissipation of block maxima during the forward process of the diffusion model (see Appendix C for proof and details). Let $\mathbf{x}_0$ be an input sample and $\mathbf{x}_n$ be the perturbed sample after $n$ iterations of the forward process, where $x_n^t = x_{n-1}^t + \epsilon_n^t$ and $\epsilon_n^t \sim \mathbb{N}(0, \sigma_{\epsilon_n^t}^2)$ is Gaussian noise. Due to the linearity of the Fourier transform operator $\mathcal{F}$, we have:

$$\mathcal{F}(\mathbf{x}_n) = \mathcal{F}(\mathbf{x}_{n-1}) + \mathcal{F}(\epsilon_n) \quad \Longrightarrow \quad f_n^k = f_{n-1}^k + \mathcal{E}_n^k \tag{5}$$

**Theorem 1.** *Under certain mild assumptions (see Appendix C), the ratio of high-frequency and low-frequency components after perturbation during the forward process of the diffusion model is:*

$$\frac{\lim_{k \to k_{\max}} |f_n^k|^2}{\lim_{k \to 0} |f_n^k|^2} = \delta \ll 1 \tag{6}$$

*where $k_{\max}$ is the index of the maximum frequency and $\delta = f_n^{k_{\max}}$, which is generally close to 0.*

In short, our findings shed light on a fundamental limitation of diffusion models while modeling block maxima and underscore the need for a more tailored approach to preserve its distribution.

## 4   Proposed Framework: FIDE

In this section, we present the detailed methodology of our proposed approach, addressing the challenges associated with capturing extreme values of time series within diffusion-based generative models. Figure 4 provides an overview of the FIDE framework.

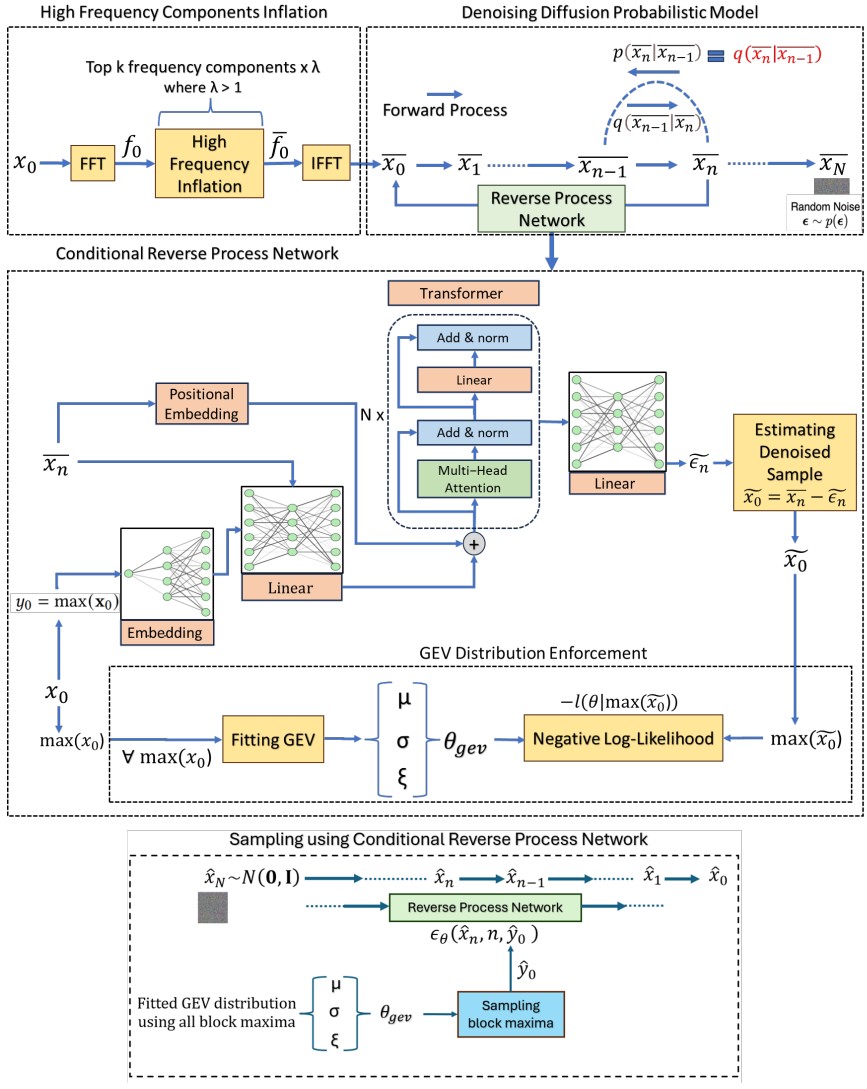

Figure 4: Proposed FIDE framework for generating time series with extreme events

## 4.1 High Frequency Components Inflation

In order to counteract the rapid decay of high-frequency components in the frequency domain while adding noise in the forward process of DDPM, we present a strategy for high-frequency inflation. Let $\mathbf{f}_0 = \mathbf{FFT}(\mathbf{x}_0)$ denote the vector of Fourier coefficients resulting from applying the discrete Fourier transform to the time series $\mathbf{x}_0$. These coefficients are arranged in ascending order from lowest to highest frequency. Consequently, the last $\kappa$ elements of $\mathbf{f}_0$ correspond to the coefficients associated with the $\kappa$ highest frequencies. Our goal is to inflate the top-$\kappa$ frequency components of $\mathbf{f}_0$ as follows:

$$\Gamma^i = \begin{cases} 1, & \text{if } i \leq \kappa \\ \gamma, & \text{if } i > T - \kappa \end{cases} \quad \text{and} \quad \bar{\mathbf{f}}_0 = \mathbf{\Gamma} \odot \mathbf{f}_0$$

where $\gamma > 1$ is the inflation weight and $\odot$ denotes the element-wise multiplication.

With the modified coefficients $\bar{\mathbf{f}}_0$, the inverse Fourier transform (IFFT) is applied to get the modified time series, $\bar{\mathbf{x}}_0 = \mathbf{IFFT}(\bar{\mathbf{f}}_0)$, containing the inflated high-frequency components. Here, the high-frequency components are inflated by $\gamma > 1$. The following theorem shows how this inflation strategy helps the high-frequency components (block maxima) diminish less rapidly in the diffusion forward process compared to before (see Appendix C for proof and details).

**Theorem 2.** *Let $\overline{f_n^k}$ be the Fourier coefficient after inflating high-frequency components with a factor of $\gamma$ such that $\gamma > 1$. Let $\delta = f_n^{k_{\max}}$ be the Fourier coefficient of the maximum frequency before inflation, and $\delta' = \delta \cdot \gamma = \overline{f_n^{k_{\max}}}$ be the Fourier coefficient of the maximum frequency after inflation. Then, using Lemma 1 and under certain mild assumptions (see Appendix C), the ratio of high-frequency and low-frequency components after inflation and perturbations is:*

$$\frac{\lim_{k \to k_{\max}} |\overline{f_n^k}|^2}{\lim_{k \to 0} |\overline{f_n^k}|^2} = \delta \cdot \gamma \tag{7}$$

Thus, by applying high-frequency inflation, the high-frequency components including abrupt block maxima will be preserved by a factor of $\gamma$ compared to the previous case. We can see the effects of this inflation empirically as well. Figure 3b shows how inflating the high-frequency components helps in preserving the block maxima values for longer iterations of the diffusion model. This enables the block maxima after high-frequency inflation to dissipate at a similar rate compared to other values in the earlier iterations. The diffusion model will have more iterations to capture the block maxima signal.

## 4.2 Forward Process

We use the inflated time series $\overline{\mathbf{x}_0}$ as input time series to be perturbed during the forward process instead of $\mathbf{x}_0$. By adopting $\overline{\mathbf{x}_0}$ as the reference for the unperturbed sample, we ensure that the denoising diffusion process takes advantage of the enhanced representation provided by the inflated high-frequency components. This nuanced adjustment contributes to the efficacy of our proposed framework in capturing and preserving essential information during the diffusion process.

## 4.3 Conditional Reverse Diffusion Process

To enable the generation of samples conditioned on block maxima, we extend the conventional diffusion model to a conditional model. Here, the reverse process is conditioned on block maxima $y_0$. Grounded in extreme value theory [4], the block maxima values $\{y_0\}$ are governed by the Generalized Extreme Value (GEV) distribution, distinctly diverging from the distribution of all values $\mathbf{x_0} \sim p_\theta(\mathbf{x}_0)$. This mandates a strategic shift in our learning objective. Rather than marginally targeting $p_\theta(\mathbf{x}_0)$, our objective now extends to mastering the joint distribution $p_\theta(\mathbf{x}_0, y_0)$, driven by a nuanced understanding of the unique characteristics inherent in extreme events and their crucial impact on the overall distribution. We formally extend the diffusion model's marginal distribution to a joint distribution in the following theorem (see Appendix C for proof and details).

**Theorem 3.** *Consider an extension of the conventional diffusion model from learning a marginal distribution $p_\theta(\mathbf{x}_0)$ to a joint distribution $p_\theta(\mathbf{x}_0, y_0)$ conditioned on block maxima $y_0$. In this context, the variational lower bound can be formulated as follows:*

$$-\log p_\theta(\mathbf{x}_0, y_0) \leq \mathbb{E}q\left[\log \frac{q(\mathbf{x}_{1:N}|\mathbf{x}_0, y_0)}{p_\theta(\mathbf{x}_{0:N})}\right] - \log p_\theta(y_0) \tag{8}$$

First, we adopt $\overline{\mathbf{x}_0}$ as the reference for the unperturbed sample $\mathbf{x}_0$ as discussed in the previous subsection. After reparameterization and ignoring the weighting term, as suggested by [12], the first term of the variation lower bound can be expressed as:

$$\mathcal{L}_{\text{DDPM}} = \mathbb{E}_{\overline{x_n}, \overline{\epsilon_n}, n, y_0} \left\| \widetilde{\epsilon}_n(\overline{\mathbf{x}_n}, n, y_0) - \overline{\epsilon_n} \right\|_2^2 \tag{9}$$

Additionally, considering a Generalized Extreme Value (GEV) distribution for block maxima, the second term is simplified as $\log \mathcal{L}_{\text{GEV}}(\mu, \sigma, \xi)$, as defined in Eq. 4.

The preceding theorem establishes a clear link between the variational lower bound and an interpretable objective loss function:

$$-\log p_\theta(\mathbf{x}_0, y_0) \leq \mathcal{L}_{\text{DDPM}} - \lambda \log \mathcal{L}_{\text{GEV}}(\mu, \sigma, \xi) := L \tag{10}$$

Here, $\mathcal{L}_{\text{DDPM}}$ represents the expected reconstruction error between actual and estimated noise, and $\log \mathcal{L}_{\text{GEV}}(\mu, \sigma, \xi)$ captures the negative log-likelihood of the block maxima governed by the GEV distribution.

### 4.4 GEV Distribution Enforcement Module

To enforce fidelity on both the block maxima and overall data distribution, we incorporate the Generalized Extreme Value (GEV) distribution within the DDPM framework following Theorem 3. We first fit a GEV distribution using maximum log-likelihood estimation with all the block maxima ($y_0$) values in the training data. The fitted GEV distribution is parameterized by $\mu$, $\sigma$, and $\xi$, denoted as $\theta_{\text{gev}} = \{\mu, \sigma, \xi\}$. Using the conditional diffusion process, the estimated noise is given by $\widetilde{\epsilon}_n(\overline{\mathbf{x}_n}, n, y_0)$. Consequently, the estimated denoised sample can be obtained as: $\widetilde{\mathbf{x}}_0 = \overline{\mathbf{x}_n} - \widetilde{\epsilon}_n$. Then, utilizing the fitted GEV distribution, the log-likelihood of the estimated denoised block maxima, $\widetilde{y}_0 = \max_{\tau \in \{1, \cdots, T\}} \widetilde{\mathbf{x}}_0^\tau$, is calculated. This negative log-likelihood, $-\log \mathcal{L}_{\text{GEV}}(\mu, \sigma, \xi, \widetilde{y}_0)$, is finally incorporated into the loss function of training.

### 4.5 Optimization

Algorithm 1 summarizes the pseudocode for training and Algorithm 2 summarizes the pseudocode for the sampling step of FIDE. The overall loss function $\mathcal{L}_{\text{FIDE}}$ is constructed by combining two key components: the DDPM loss $\mathcal{L}_{\text{DDPM}}$ and the negative log-likelihood of the Generalized Extreme Value (GEV) distribution $-\mathcal{L}_{\text{GEV}}$. The formulation is expressed as follows:

$$\mathcal{L}_{\text{FIDE}} = \mathbb{E}_{\overline{\mathbf{x}_n}, \overline{\epsilon_n}, n, y_0} \|\widetilde{\epsilon}_n(\overline{\mathbf{x}_n}, n, y_0) - \overline{\epsilon_n}\|_2^2 - \lambda \cdot \log \mathcal{L}_{\text{GEV}}(\mu, \sigma, \xi, \widetilde{y}_0) \tag{11}$$

where $\lambda$ is a hyperparameter controlling the influence of the GEV distribution on the loss.

In this context, $\mathcal{L}_{\text{DDPM}}$ evaluates the mean squared difference between the estimated noise term $\widetilde{\epsilon}_n$ and the true noise term $\epsilon_n$ within the conditional diffusion process. Its purpose is to guide the generative model towards effectively capturing the underlying data distribution. The second element, $-\log \mathcal{L}_{\text{GEV}}(\mu, \sigma, \xi, \widetilde{y}_0)$, encapsulates the negative log-likelihood of the GEV distribution. This component assesses how well the fitted GEV distribution aligns with the estimated block maxima values $\widetilde{y}_0$ derived from the denoised samples. Here, the log-likelihood has a negative sign to indicate a minimization objective, aligning with the overall goal of minimizing the loss function.

---

**Algorithm 1** Training

**repeat**
    $\mathbf{x}_0 \sim q(\mathbf{x}_0)$ where $\mathbf{x}_0 = (x_0^1, x_0^2, \cdots, x_0^T)$
    $\mathbf{f}_0 = \text{FFT}(\mathbf{x_0})$
    $\overline{\mathbf{f}_0} = \gamma \odot \mathbf{f_0}$
    $\overline{\mathbf{x}_0} = \text{IFFT}(\overline{\mathbf{f}_0})$
    $n \sim \text{Uniform}(\{1, \cdots, N\})$
    $\overline{\epsilon_n} \sim \mathcal{N}(0, \mathbf{I})$
    $y_0 = \max(\mathbf{x}_0)$
    $\overline{\mathbf{x}_n} = \sqrt{\alpha_n}\, \overline{\mathbf{x}_0} + \sqrt{1 - \alpha_n}\, \overline{\epsilon_n}$
    $\widetilde{\mathbf{y}_0} = \max(\overline{\mathbf{x}_n} - \widetilde{\epsilon}_n(\overline{\mathbf{x}_n}, n, y_0))$
    Take the gradient step on
        $\nabla_\theta \|\widetilde{\epsilon}_n(\overline{\mathbf{x}_n}, n, y_0)) - \overline{\epsilon_n}\|_2^2$
            $-\lambda \cdot \log \mathcal{L}_{\text{GEV}}(\mu, \sigma, \xi, \widetilde{y}_0)$
**until** converged

---

**Algorithm 2** Sampling

**Input:** Block maxima $\hat{y}_0 \sim \text{GEV}(y_0)$ and Trained Model $\widetilde{\epsilon}$
**Output:** Generate time series, $\hat{\mathbf{x}}_0$.

$\hat{\mathbf{x}}_N \sim \mathcal{N}(0, \sigma^2)$
**for** $n = N, \cdots, 1$ **do**
    $\mathbf{z} \sim \mathcal{N}(0, \mathbf{I})$
    $\hat{\mathbf{x}}_{n-1} = \frac{1}{\sqrt{\alpha_n}}(\hat{\mathbf{x}}_n - \frac{1-\alpha_n}{\sqrt{1-\alpha_n}} \widetilde{\epsilon}_n(\hat{\mathbf{x}}_n, n, \hat{y}_0))$
            $+ \sigma_n \mathbf{z}$
**end for**
**return** $\hat{\mathbf{x}}_0$

---

## 5 Experimental Evaluation

We have performed extensive experiments to evaluate the performance of our FIDE framework. All the code and datasets used in this paper are available at `https://github.com/galib19/FIDE`. The datasets used are described in Appendix D.

We compared our proposed framework against various generative models: **(1) GAN-based:** We utilize two GAN-based approaches as our baselines. The first approach is Conditional GAN (cGAN [15]), which introduces conditional information to the training process, enabling targeted generation based on specified conditions. The second baseline is TimeGAN [21], which is a generative model designed specifically for time-series generation. **(2) VAE-based:** We employ beta-VAE [11], conditional beta-VAE [16], and TimeVAE [6] as baseline methods for comparison. Both beta-VAE and conditional

**Table 1** Comparison of generated samples' block maxima distribution metrics and predictive score using the various methods. **Bold** and Underlined entries denote the best and second-best result

| Metrics | Methods | AR1 | Stock | Energy | Temperature | ECG |
|---|---|---|---|---|---|---|
| JS Divergence | beta-VAE | 0.0211±0.0187 | 0.1105±0.0188 | 0.0722±0.0095 | 0.0140±0.0125 | 0.1210±0.0214 |
| | c-beta-VAE | 0.0190±0.0125 | 0.1011±0.0152 | 0.0710±0.0088 | 0.0109±0.0098 | 0.1120±0.0352 |
| | TimeVAE | 0.0015±0.0003 | 0.1054±0.0071 | 0.0795±0.0085 | 0.0096±0.0002 | 0.0985±0.0078 |
| | TimeGAN | 0.0840±0.0109 | 0.1411±0.1585 | 0.0950±0.0089 | 0.0112±0.0012 | 0.1620±0.0221 |
| | cGAN | 0.0690±0.0091 | 0.1211±0.0205 | 0.0890±0.0093 | 0.0091±0.0008 | 0.1440±0.0211 |
| | RealNVP | 0.0754±0.0121 | 0.1185±0.0108 | 0.0905±0.0084 | 0.0089±0.0007 | 0.1411±0.0116 |
| | Fourier-Flows | 0.0612±0.0045 | 0.1108±0.0195 | 0.0820±0.0044 | 0.0078±0.0010 | 0.1398±0.0202 |
| | DDPM | 0.0010±0.0007 | 0.0912±0.0062 | 0.0752±0.0082 | 0.0082±0.0009 | 0.1041±0.0122 |
| | Diffusion-TS | 0.0011±0.0008 | 0.0854±0.0045 | 0.0712±0.0071 | 0.0077±0.0008 | 0.1005±0.0108 |
| | FIDE (Ours) | **0.0004±0.0001** | **0.0700±0.0061** | **0.0680±0.0092** | **0.0007±0.0001** | **0.0930±0.0082** |
| KL Divergence | beta-VAE | 0.0110±0.0024 | 0.1947±0.0184 | 0.1210±0.0146 | 0.0410±0.0128 | 0.2020±0.0048 |
| | c-beta-VAE | 0.0091±0.0012 | 0.1744±0.0105 | 0.1160±0.0174 | 0.0360±0.0114 | 0.1880±0.0079 |
| | TimeVAE | 0.0105±0.0007 | 0.2514±0.0152 | 0.1625±0.0095 | 0.0490±0.0006 | 0.2254±0.0068 |
| | TimeGAN | 0.1920±0.0156 | 0.2425±0.0251 | 0.1590±0.0198 | 0.0550±0.0145 | 0.2540±0.0254 |
| | cGAN | 0.1240±0.0122 | 0.2101±0.0115 | 0.1510±0.0211 | 0.0490±0.0125 | 0.2210±0.0184 |
| | RealNVP | 0.1298±0.0215 | 0.2295±0.0154 | 0.1605±0.0310 | 0.0512±0.0108 | 0.2305±0.0145 |
| | Fourier-Flows | 0.1235±0.0104 | 0.2045±0.0255 | 0.1458±0.0345 | 0.0505±0.0136 | 0.2254±0.0141 |
| | DDPM | 0.0062±0.0008 | 0.1915±0.0125 | 0.1120±0.0108 | 0.0326±0.0090 | 0.1905±0.0094 |
| | Diffusion-TS | 0.0054±0.0007 | 0.1889±0.0108 | 0.1089±0.0115 | 0.0311±0.0078 | 0.1894±0.0081 |
| | FIDE (Ours) | **0.0030±0.0009** | **0.1504±0.0128** | **0.0950±0.0098** | **0.0029±0.0008** | **0.1810±0.0084** |
| CRPS | beta-VAE | 0.1247±0.0189 | 0.3149±0.0348 | 0.2410±0.0298 | 0.1554±0.0214 | 0.3059±0.0454 |
| | c-beta-VAE | 0.1154±0.0151 | 0.2698±0.0214 | 0.2574±0.0241 | 0.1420±0.0311 | 0.3150±0.0414 |
| | TimeVAE | 0.1511±0.0081 | 0.2547±0.0155 | 0.2853±0.1082 | 0.1847±0.0071 | 0.3252±0.0204 |
| | TimeGAN | 0.1858±0.0214 | 0.2825±0.0418 | 0.2685±0.0284 | 0.2110±0.0287 | 0.3240±0.0401 |
| | cGAN | 0.1224±0.0157 | 0.2689±0.0301 | 0.2385±0.0187 | 0.1990±0.0214 | 0.2985±0.0311 |
| | RealNVP | 0.1325±0.0144 | 0.2545±0.0258 | 0.2541±0.0214 | 0.2014±0.0354 | 0.2824±0.0425 |
| | Fourier-Flows | 0.1305±0.0254 | 0.2589±0.0214 | 0.2415±0.0211 | 0.1975±0.0251 | 0.2884±0.0215 |
| | DDPM | 0.0422±0.0084 | 0.2422±0.0187 | 0.2199±0.0874 | 0.1516±0.0211 | 0.2488±0.0388 |
| | Diffusion-TS | 0.0398±0.0092 | 0.2358±0.0211 | 0.2125±0.0454 | 0.1525±0.0315 | 0.2415±0.0451 |
| | FIDE (Ours) | **0.0310±0.0098** | **0.2115±0.0152** | **0.2085±0.0985** | **0.0517±0.0082** | **0.2345±0.0204** |
| Predictive Score | beta-VAE | 0.6350±0.0201 | 0.9528±0.0314 | 0.7410±0.0187 | 0.6814±0.0108 | 0.9420±0.0142 |
| | c-beta-VAE | 0.6240±0.0145 | 0.9226±0.0165 | 0.7317±0.0163 | 0.6718±0.0025 | 0.9310±0.0214 |
| | TimeVAE | 0.6150±0.0104 | 0.9140±0.0218 | 0.7325±0.0195 | 0.6723±0.0036 | 0.9150±0.0112 |
| | TimeGAN | **0.6050±0.0104** | 0.8950±0.0198 | 0.7280±0.0187 | 0.6718±0.0047 | **0.8960±0.0084** |
| | cGAN | 0.6120±0.0014 | 0.9354±0.0210 | 0.7310±0.0147 | 0.6847±0.0041 | 0.9220±0.0191 |
| | RealNVP | 0.6884±0.0011 | 0.9988±0.0354 | 0.7898±0.0254 | 0.7852±0.0017 | 0.9730±0.0215 |
| | Fourier-Flows | 0.6925±0.0031 | 0.9844±0.0241 | 0.7955±0.0088 | 0.7871±0.0021 | 0.9655±0.0221 |
| | DDPM | 0.6148±0.0081 | 0.8997±0.0111 | 0.7350±0.0102 | 0.6708±0.0098 | 0.9121±0.0121 |
| | Diffusion-TS | 0.6105±0.0045 | 0.8912±0.0105 | 0.7355±0.0084 | 0.6708±0.0108 | 0.9089±0.0095 |
| | FIDE (Ours) | 0.6081±0.0098 | **0.8871±0.0104** | **0.7240±0.0087** | **0.6694±0.0082** | 0.9040±0.0112 |

beta-VAE incorporate a specific disentanglement objective to encourage the model to learn more interpretable and factorized representations while TimeVAE [6] promotes interpretability. **(3) Flow-based:** We use normalizing flows-based approaches such as RealNVP [7] and Fourier-Flows [1] as our baseline methods. **(4) Diffusion-based:** We consider two baselines for comparison, namely, the denoising diffusion probabilistic model (DDPM) [12] and time series diffusion model called Diffusion-TS [22].

## 5.1 Experimental Settings

We partitioned each dataset into training, validation, and testing, according to a 8:1:1 ratio. We repeated the experiments 5 times. Prior to applying the various algorithms, the time series data is standardized to have zero mean and unit variance. The encoder component of our framework employs a 3-layer transformer architecture, accompanied by fully connected layers. The training was facilitated using the Adam optimizer. For all the methods, we perform extensive hyperparameter tuning on the length of the embedding vector, the number of hidden layers, the number of nodes, the learning rate, and the batch size. The optimal hyperparameters were determined using the Ray Tune framework, integrating an Asynchronous Successive Halving Algorithm (ASHA) scheduler to enable early stopping. All experiments were conducted on NVIDIA T4 GPU.

To assess the effectiveness of the proposed framework, we utilize four metrics: Jensen-Shannon (JS) Divergence, KL Divergence, CRPS (Continuous Rank Probability Score), and Predictive Score. The first three metrics examine how well the generated samples fit the original data distribution. The

**Table 2** Comparison of the generated sample distribution for all values using KL divergence and CRPS metrics. **Bold** and Underlined entries denote the best and second-best result. For the JS Divergence and Predictive Score metrics, the results are given in Appendix E.

| Metrics | Methods | AR1 | Stock | Energy | Temperature | ECG |
|---|---|---|---|---|---|---|
| KL Divergence | beta-VAE | 0.0020±0.0003 | 0.0188±0.0016 | 0.0181±0.0015 | 0.0025±0.0003 | 0.0031±0.0004 |
| | c-beta-VAE | 0.0017±0.0004 | 0.0178±0.0019 | 0.0177±0.0017 | 0.0022±0.0004 | 0.0028±0.0004 |
| | TimeVAE | 0.0016±0.0003 | 0.0169±0.0015 | 0.0159±0.0021 | 0.0018±0.0002 | 0.0026±0.0003 |
| | TimeGAN | 0.0025±0.0003 | 0.0182±0.0025 | 0.0161±0.0016 | 0.0021±0.0005 | 0.0034±0.0005 |
| | cGAN | 0.0018±0.0003 | 0.0178±0.0018 | 0.0169±0.0016 | 0.0015±0.0003 | 0.0029±0.0003 |
| | RealNVP | 0.0023±0.0004 | 0.0185±0.0019 | 0.0185±0.0017 | 0.0019±0.0004 | 0.0036±0.0002 |
| | Fourier-Flows | 0.0019±0.0003 | 0.0173±0.0021 | 0.0165±0.0015 | 0.0017±0.0003 | 0.0028±0.0003 |
| | DDPM | **0.0010±0.0001** | 0.0117±0.0011 | 0.0114±0.0010 | **0.0009±0.0001** | **0.0019±0.0003** |
| | Diffusion-TS | 0.0011±0.0001 | **0.0114±0.0012** | 0.0116±0.0009 | 0.0010±0.0002 | **0.0019±0.0003** |
| | FIDE (Ours) | 0.0012±0.0001 | 0.0121±0.0015 | **0.0109±0.0009** | 0.0011±0.0002 | 0.0021±0.0004 |
| CRPS | beta-VAE | 0.0201±0.0041 | 0.4955±0.0125 | 0.4985±0.0102 | 0.0914±0.0010 | 0.1425±0.0049 |
| | c-beta-VAE | 0.1984±0.0022 | 0.4205±0.0148 | 0.4514±0.0210 | 0.0899±0.0009 | 0.1388±0.0068 |
| | TimeVAE | 0.1848±0.0038 | 0.4841±0.085 | 0.4815±0.0189 | 0.0889±0.0009 | 0.1422±0.0077 |
| | TimeGAN | 0.2412±0.0019 | 0.3941±0.0115 | 0.4415±0.0171 | 0.0911±0.0008 | 0.1262±0.0062 |
| | cGAN | 0.1974±0.0012 | 0.4451±0.0201 | 0.3914±0.0211 | 0.0903±0.0007 | 0.1298±0.0056 |
| | RealNVP | 0.2511±0.0019 | 0.4254±0.0194 | 0.5125±0.0184 | 0.0919±0.0007 | 0.1405±0.0035 |
| | Fourier-Flows | 0.2214±0.0024 | 0.3814±0.0164 | 0.4514±0.0123 | 0.0912±0.0008 | 0.1281±0.0077 |
| | DDPM | 0.1595±0.0018 | **0.2955±0.0144** | **0.3215±0.0154** | 0.0875±0.0006 | 0.1028±0.0062 |
| | Diffusion-TS | 0.1565±0.0016 | 0.2985±0.0174 | 0.3285±0.0149 | **0.0863±0.0007** | **0.1018±0.0045** |
| | FIDE (Ours) | **0.1541±0.0021** | 0.3001±0.0191 | 0.3251±0.0177 | 0.0893±0.0007 | 0.1061±0.0054 |

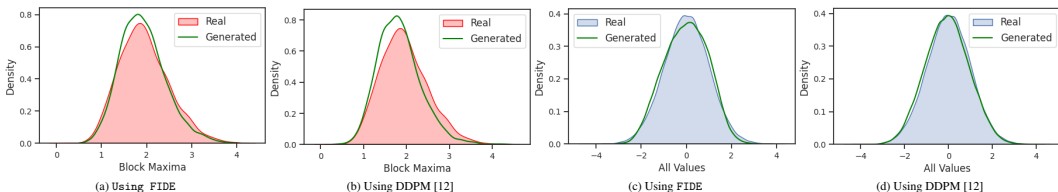

Figure 5: Comparison of block maxima distribution and all values distribution for real and generated samples using the proposed `FIDE` model and DDPM [12] when applied to the synthetic AR(1) dataset.

fourth metric, Predictive Score [21], evaluates the generative model's ability to replicate the temporal characteristics of the original data. This is done by training an LSTM-based sequence model for time series forecasting using the synthetic samples produced by each generative model. The model's performance is measured by its mean absolute error (MAE) on the original test data, providing insight into how well the generative model preserves the temporal patterns of the data. In short, the evaluation focuses on forecasting block maxima on the test dataset using the model trained on generated data.

## 5.2 Experimental Results

Table 1 compares the performance of `FIDE` against the various baselines in terms of their ability to capture the block maxima distribution for 5 diverse datasets (AR1, Stock, Energy, Temperature, and ECG). In terms of the distribution metrics (JS divergence, KL divergence, and CRPS), `FIDE` consistently achieves the best results, providing evidence of `FIDE`'s superior performance in preserving the block maxima distribution. For the Predictive Score metric, `FIDE` achieves the best results in 3 out of 5 datasets and ranks second in the remaining 2 datasets. To further illustrate `FIDE`'s capabilities, Figures 5-(a) and (b) compare the distribution of block maxima values generated by DDPM [12] and `FIDE` for the AR(1) dataset. Note that, while DDPM struggles to capture the block maxima distribution accurately, `FIDE` generates samples that more faithfully preserve the fidelity of the distribution. This improvement is particularly noticeable in the upper tail behavior, which is critical for applications that require precise modeling of extreme block maxima values. This superior performance is not surprising as it directly results from our method's emphasis on block maxima distribution, achieved through the introduction of frequency inflation, conditional generation based on block maxima, and incorporation of the GEV distribution into the generative modeling framework.

As `FIDE` prioritizes the accurate modeling of block maxima, we have also evaluated its efficacy in capturing the distribution of all (block maxima and non-block maxima) values in time series. The

**Table 3** Ablation Study of generated samples' block maxima distribution metrics and predictive score using the proposed `FIDE` model and without individual component of the model

| Metrics | Methods | AR1 | Stock | Energy | Temperature | ECG |
|---|---|---|---|---|---|---|
| Jensen–Shannon Divergence | FIDE - frequency inflation | 0.0006+0.0001 | 0.0898+0.0054 | 0.0822+0.0084 | 0.0009+0.0001 | 0.0984+0.0058 |
| | FIDE - conditional | 0.0007+0.0001 | 0.1054+0.0089 | 0.0941+0.0098 | 0.0010+0.0002 | 0.1102+0.0098 |
| | FIDE - GEV Loss | 0.0005+0.0001 | 0.0813+0.0035 | 0.0715+0.0041 | 0.0008+0.0001 | 0.0922+0.0056 |
| | FIDE | **0.0004+0.0001** | **0.0700+0.0061** | **0.0680+0.0092** | **0.0007+0.0001** | **0.0930+0.0082** |
| KL Divergence | FIDE - frequency inflation | 0.0042+0.0008 | 0.1559+0.0161 | 0.1054+0.0049 | 0.0036+0.0006 | 0.1854+0.0064 |
| | FIDE - conditional | 0.0051+0.0010 | 0.1689+0.0210 | 0.1089+0.0095 | 0.0041+0.0010 | 0.1901+0.0063 |
| | FIDE - GEV Loss | 0.0039+0.0007 | 0.1551+0.0188 | 0.1021+0.0088 | 0.0032+0.0009 | 0.1823+0.0092 |
| | FIDE | **0.0030+0.0009** | **0.1504+0.0128** | **0.0950+0.0098** | **0.0029+0.0008** | **0.1810+0.0084** |
| CRPS | FIDE - frequency inflation | 0.0391+0.0078 | 0.2172+0.0158 | 0.2152+0.0791 | 0.0649+0.0081 | 0.2372+0.0181 |
| | FIDE - conditional | 0.0335+0.0089 | 0.2165+0.0132 | **0.2082+0.0768** | 0.0651+0.0047 | 0.2382+0.0184 |
| | FIDE - GEV Loss | 0.0415+0.0087 | 0.2232+0.0203 | 0.2189+0.0874 | 0.0815+0.0104 | 0.2456+0.0399 |
| | FIDE | **0.0310+0.0098** | **0.2115+0.0152** | 0.2085+0.0985 | **0.0517+0.0082** | **0.2345+0.0204** |
| Predictive Score | FIDE - frequency inflation | **0.6070+0.0112** | 0.8942+0.0158 | 0.7264+0.0069 | 0.6711+0.0091 | 0.9081+0.0154 |
| | FIDE - conditional | 0.6095+0.0079 | 0.8901+0.0141 | 0.7261+0.0081 | 0.6715+0.0078 | 0.9059+0.0122 |
| | FIDE - GEV Loss | 0.6089+0.0089 | 0.8891+0.0122 | 0.7269+0.0074 | 0.6712+0.0009 | 0.9062+0.0058 |
| | FIDE | 0.6081+0.0098 | **0.8871+0.0104** | **0.7240+0.0087** | **0.6694+0.0082** | **0.9040+0.0112** |

results are shown in Tables 2 and 4 (in Appendix E). Note that `FIDE` achieves comparable performance to state-of-the-art methods like DDPM [12] and Diffusion-TS [22]. This is further illustrated by the distribution plots of all values for DDPM and `FIDE` given in Figure 5-(c) and (d). The results in Table 2 also show that `FIDE` consistently outperforms VAE-based, GAN-based, and Flow-based alternatives. For Predictive Score, while TimeGAN and TimeVAE show marginally better results, `FIDE` maintains competitive performance against other baseline methods. These results suggest minimal performance degradation when applying `FIDE` to time series data. Despite its emphasis on block maxima values, this does not significantly compromise its ability to model the overall distribution. This positions `FIDE` as a robust and versatile generative model for capturing extreme values in time series.

## 5.3   Ablation Study

In our ablation study depicted in Table 3, we systematically assessed the individual contributions of each component within our proposed framework. By selectively deactivating elements such as the GEV loss, conditional block maxima input, and high-frequency inflation module, we observed consistent performance degradation across all scenarios. Notably, the absence of the conditional block maxima input significantly impacted the Jenson-Shannon Divergence and KL Divergence metrics, while the lack of the GEV loss had the most pronounced effect on the CRPS metric. Surprisingly, the predictive score remained relatively resilient to the deactivation of any single component, suggesting a degree of redundancy or compensatory mechanisms among the remaining components. Overall, our ablation study highlights the indispensable role of each component in achieving optimal performance in our model. In summary, our findings underscore the holistic importance of the individual components, with their synergistic interplay contributing to the overall effectiveness of `FIDE`.

## 6   Conclusions

This framework examines the challenges of applying diffusion models to capture extreme values in time series. Through a comprehensive exploration of the constraints within current diffusion-based models, the proposed `FIDE` framework addresses these limitations by introducing a novel strategy to maintain high-frequency components of the time series. `FIDE` extends conventional diffusion models to enable conditional generation of block maxima by integrating a loss function based on the generalized extreme value (GEV) distribution. The superiority of the framework over various baseline methods is validated through rigorous experiments on both synthetic and real-world data.

## 7   Acknowledgment

This research is supported by the U.S. National Science Foundation under grant IIS-2006633. Any use of trade, firm, or product names are for descriptive purposes only and do not imply endorsement by the U.S. Government.

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

# A  Related Works

Time series generation has been a subject of extensive research, leveraging a variety of statistical [23] and machine-learning [16, 21] techniques to capture temporal dependencies and complexities within data. Generative methods, including Generative Adversarial Networks (GANs) [10], Variational Autoencoders (VAEs) [13], normalizing flows [18], and diffusion-based approaches [12, 19], have demonstrated efficacy in time series generation and garnered interest due to their ability to learn underlying data distributions for data generation. Normalizing flows are constrained by their computational complexity, limited expressiveness, and suboptimal sample quality, thereby restricting their capacity for effective modeling. Numerous works have delved into enhancing GANs, introducing variations like RcGAN [15] and TimeGAN [21], which have demonstrated improvements in generating realistic time series data. TimeGAN [21] specifically adopts a GAN architecture to generate time-series data, employing an encoder and decoder to transform a time-series sample into latent vectors. However, GAN-based generative models are susceptible to issues like mode collapse and unstable behavior during training. While VAEs have not been extensively applied to synthetic time series generation, their effectiveness in addressing related challenges, such as time series imputation [8], suggests their potential utility in this domain. Diffusion-based models are also gaining traction for their ability to generate high-quality data such as images and videos, bypassing the challenges associated with discriminator networks in GANs and avoiding the artifact-prone lower-dimensional latent spaces of VAEs. There are a couple of diffusion-based works [20, 2] that have been employed for time series, but they are specifically designed for discriminative tasks.

While generative AI for time series offers numerous advantages, it has not been extensively explored, especially in terms of modeling extreme values. The difficulty of modeling extremes using generative models such as normalizing flows [14] has been recognized in previous research. Studies by Wiese et al. (2019) and Jaini et al. (2020) highlight the inability of normalizing flows to accurately capture heavy-tailed marginal distributions. Specifically, these studies show that any attempt to map heavy-tailed distributions to light-tailed distributions (e.g., Gaussian) cannot maintain Lipschitz-boundedness. However, this challenge remains largely unexplored within the realm of diffusion models.

# B  Relationship between Abrupt Block Maxima and High Frequency Components

This section presents the relationship between abrupt block maxima and high frequency components of a time series.

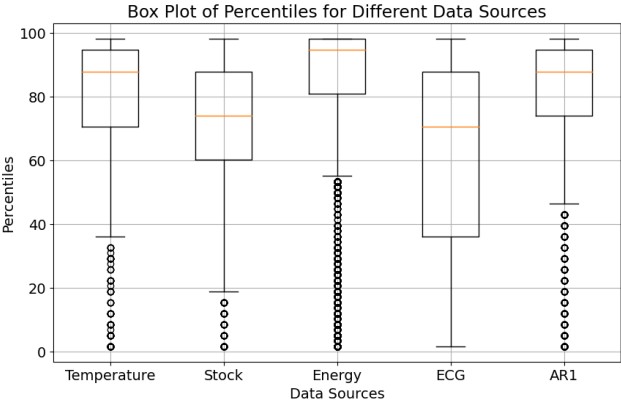

Figure 6: Percentile distribution of first order derivatives for the block maxima values in different time series datasets. Observe that the derivatives tend to exhibit elevated percentile values.

**Definition 1** (Abrupt Block Maxima). *Let,* $\mathbf{x}_0 \in \mathbb{R}^T$ *be a time series of length* $T$ *and* $y_0 \equiv x_0^\tau = \max_{t \in 1,...,T} x_0^t$ *be its block maxima, where* $\tau$ *is the time step of the block maxima. Then,* $x_0^\tau$ *is considered an abrupt block maxima if* $\frac{dx}{dt}|_{t=\tau} > \rho$*, where* $\rho$ *is a threshold.*

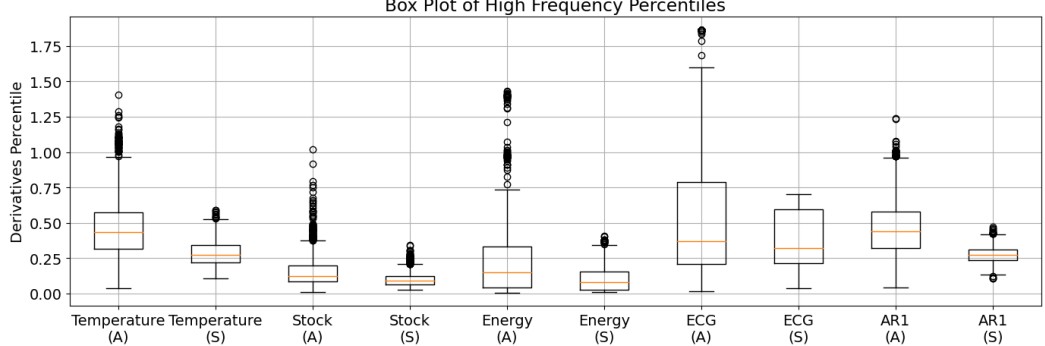

Figure 7: The summation of high frequency terms for abrupt (A) changes is consistently higher than for smooth (S) changes. (A) denotes abrupt changes, (S) denotes smooth changes.

We argue that the block maxima values in real-world time series often exhibit an abrupt change behavior compared to the non-block maxima values. To substantiate this, we conduct an empirical analysis across five distinct datasets, wherein we assess the percentile distribution of the first-order derivatives associated with the block maxima values, as depicted in Figure 6. Specifically, given a time series, we first partition it into a set of disjoint time windows and compute the block maxima value within each window along with the first-order difference, $\Delta x^t = x_0^t - x_0^{t-1}$, for each time step $t$. We then compute the percentile of $\Delta x^\tau$ associated with the block maxima $x_0^\tau$ of the window and plot its distribution using a boxplot as shown in Figure 6. Our findings affirm the conjecture that the block maxima tends to exhibit an elevated value for its time derivative (with a median larger than 70% of all first-order differences), thereby indicating a notable association between block maxima occurrences and the abrupt changes in a time series.

More importantly, the abrupt changes are strongly influenced by the high frequency components of the time series. This can be observed by differentiating the inverse Fourier transform shown in Equation 2 and decomposing the derivative into low and high frequency components:

$$\frac{dx_{m,0}^t}{dt} = \frac{1}{T} \sum_{k=1}^{T} f_{m,0}^k \frac{d}{dt}\left[e^{i2\pi tk/T}\right] = \frac{1}{T} \sum_{k=1}^{T} \frac{i2\pi k}{T} f_{m,0}^k e^{i2\pi tk/T} = \frac{1}{T} \sum_{k=1}^{T} i\omega_k f_{m,0}^k e^{i\omega_k t},$$

where $\omega_k = \frac{2\pi k}{T}$. Let $\kappa$ be the threshold index for dividing the frequencies into low ($k \leq \kappa$) and high ($k > \kappa$) frequency components. Then, we have:

$$\frac{dx_{m,0}^t}{dt} = \frac{1}{T} \sum_{l=1}^{\kappa} i\omega_l f_{m,0}^l e^{i\omega_l t} + \frac{1}{T} \sum_{h=\kappa+1}^{T} i\omega_h f_{m,0}^h e^{i\omega_h t} \tag{12}$$

To illustrate the impact of high frequency components on the abrupt changes, we compute the value of the second term in Equation (7) for time steps with abrupt[1] (A) and non-abrupt (S) changes for various datasets. The results shown in Fig. 7 suggest that the sum of high frequency terms for abrupt changes is consistently higher than the sum of high frequency terms for non-abrupt changes. In essence, the abrupt block maxima values often manifest as high-frequency components in the Fourier domain as they introduce sharp transitions in the time domain signal. This explains the high residual shown in Fig. 2 for the block maxima when the high frequency components are zeroed out.

## C  Theoretical Analysis

Let $\epsilon_n^t$ be the Gaussian noise added in the diffusion step $n$ during the forward process of the diffusion model and $x_0^t$ be the original value in the time series. The perturbed signal at diffusion step $n$ can be expressed as: $x_n^t = x_{n-1}^t + \epsilon_n^t$. Let $\sigma_n^2$ and $\sigma_{\epsilon_n}^2$ be the variances of perturbed time series and noise respectively at diffusion step $n$ while $\sigma_{n-1}^2$ is the variance of the time series at diffusion step $n-1$.

---

[1]A time step has abrupt change if its $\frac{dx}{dt}$ is in the top 90%.

Note that $\sigma^2_{\epsilon_n}$ increases linearly according to a linear noise scheduler as the diffusion step increases. Let $\mathcal{S}^f_n(\omega_k)$ and $\mathcal{S}^{\mathcal{E}}_n(\omega_k)$ denote the power spectral density (PSD) for the $k$-th frequency component of the perturbed time series and noise respectively for the diffusion step $n$, where $\omega_k = \frac{2\pi k}{T}$.

Our theoretical analysis is based on the following assumptions:

**Assumption 1.** *An abrupt block maxima is linked to the high-frequency components of the time series.*

**Assumption 2.** *The noise $\epsilon^t_n$ is a stationary random process with a constant power spectral density (PSD) for diffusion step $n$, i.e., $\forall k : \mathcal{S}^{\mathcal{E}}_n(\omega_k) \approx \sigma^2_{\epsilon_n}$.*

**Assumption 3.** *The power spectral density (PSD) of the perturbed time series for diffusion step $n$ can be modeled using the following generalized Gaussian function:*

$$\mathcal{S}^f_n(\omega_k) = \sigma^2_n \cdot \exp(-\alpha_n |\omega_k|^{\beta_n}) \tag{13}$$

*where $\alpha_n$ is a scaling factor, and $\beta_n$ is a shape parameter.*

**Remark 1.** *The rationale and supporting evidence for Assumption 1 is presented in Section 3.*

**Remark 2.** *Assumption 2 is intuitive as the Gaussian noise used in the diffusion model has approximately constant PSD over the frequency range of interest.*

**Remark 3.** *Assumption 3 is reasonable as for most real-world time series, the energy spectrum is localized at the lower frequency ($\omega_k \approx 0$), also known as the fundamental frequency, which quickly decays with increasing frequency ($\omega_k \to \omega_{max}$) [5]. Therefore, we use the generalized Gaussian function to model this decaying behavior. Note that, the exponential decay behavior will eventually transition into a uniform distribution according to the diffusion model's forward process. Consequently, the shape $\beta_n$ of the distribution function (Eqn 13) is not constant; instead, it evolves with the diffusion step $n$ to remain consistent with the diffusion model's forward process. Initially, when $n$ is small, $1 \leq \beta_n \leq 2$, representing an exponential decay. However, as $n \to N$, where $N$ is the final diffusion step, $\beta_n \to \infty$, and the PSD transitions to a uniform distribution. This transformation occurs because, according to the forward process of the diffusion model, as $n$ increases, Gaussian noise with linearly increasing variance is added to the perturbed time series, making the signal increasingly noise-like until it becomes white noise at step $N$.*

**Lemma 1.** *The difference between the variance of the perturbed time series $x^t_n$ and the variance of the noise $\epsilon^t_n$ is equal to a constant $\zeta$ such that:*

$$\sigma^2_n - \sigma^2_{\epsilon_n} = \zeta \tag{14}$$

*where $\zeta = \sigma^2_0 + \sum_{i=1}^{n-1} \sigma^2_{\epsilon_i}$*

*Proof.* First, we have the following:

$$x^t_n = x^t_{n-1} + \epsilon^t_n \tag{15}$$

Applying the variance operator to both sides of Equation 15 and utilizing the property that the variance of a sum of independent random variables is the sum of their individual variances, we obtain:

$$\sigma^2_n = \sigma^2_{n-1} + \sigma^2_{\epsilon_n} \tag{16}$$

where $\sigma^2_{\epsilon_n}$ denotes the variance of the noise at step $n$. Recursively applying Equation 16, we can express the variance of the perturbed time series at step $n$ as:

$$\sigma^2_n = \sigma^2_0 + \sum_{i=1}^{n} \sigma^2_{\epsilon_i} \tag{17}$$

Subtracting $\sigma^2_{\epsilon_n}$ from both sides of Equation 17, we get:

$$\sigma^2_n - \sigma^2_{\epsilon_n} = \sigma^2_0 + \sum_{i=1}^{n-1} \sigma^2_{\epsilon_i} \tag{18}$$

Therefore, the difference between the variance of the perturbed time series and the variance of the noise at step $n$ is equal to the constant $\zeta = \sum_{i=1}^{n-1} \sigma^2_{\epsilon_i}$, which completes the proof. $\qquad \square$

The lemma establishes a crucial bound on the difference between the variances of the perturbed time series and the noise, which is leveraged in the subsequent theorem to analyze the behavior of the Fourier transform of the perturbed time series at low and high frequencies.

We now provide the proof for Theorem 1 in the main paper.

**Proof for Theorem 1**  Using Lemma 1 and Assumptions 1, 2, and 3, we can prove the theorem as follows: For low frequencies, i.e., $\omega_k \to 0$, taking the limit as $\omega_k \to 0$ on the expression for the signal power spectral density $\mathcal{S}_n^f(\omega)$, we have:

$$\lim_{k \to 0} \mathcal{S}_n^f(\omega_k) = \lim_{k \to 0} \sigma_n^2 \cdot \exp(-\alpha_n |\omega_k|^{\beta_n}) = \sigma_n^2$$

since $\exp(-\alpha_n |0|^{\beta_n}) = \exp(0) = 1$. Therefore, as $k \to 0$, $\mathcal{S}_n^f(\omega_k) = |f_n^k|^2$ approaches the variance $\sigma_n^2$. So, we can write:

$$\lim_{k \to 0} \mathcal{S}_n^f(\omega_k) = \lim_{k \to 0} |f_n^k|^2 = \sigma_n^2 = \zeta + \sigma_{\epsilon_n}^2 = \sigma_0^2 + \sum_{i=1}^{n-1} \sigma_{\epsilon_i}^2 + \sigma_{\epsilon_n}^2 = \sigma_0^2 + \sum_{i=1}^{n} \sigma_{\epsilon_i}^2$$

Similarly, for high frequencies, i.e., $k \to k_{\max}$, taking the limit as $\omega \to \omega_{\max}$ on the expression for the signal power spectral density $\mathcal{S}_n^f(\omega_k)$ (Eq. (13)), we have:

$$\lim_{k \to k_{\max}} \mathcal{S}_n^f(\omega_k) = \lim_{k \to k_{\max}} \sigma_n^2 \cdot \exp(-\alpha_n |\omega|^{\beta_n}) \to \sigma_n^2 \cdot \delta$$

where $\delta = f_n^{k_{\max}} \ll 1$.

As $k \to k_{\max}$, $\mathcal{S}_n^f(\omega_k)$ approaches $\sigma_n^2 \cdot \delta$. We can also write:

$$\lim_{k \to k_{\max}} |f_n^k|^2 = \delta \cdot \left(\zeta + \sigma_{\epsilon_n}^2\right) = \delta \cdot \left(\sigma_0^2 + \sum_{i=1}^{n} \sigma_{\epsilon_i}^2\right)$$

Taking the ratio of high-frequency and low-frequency components after perturbations yields:

$$\frac{\lim_{k \to k_{\max}} |f_n^k|^2}{\lim_{k \to 0} |f_n^k|^2} = \delta$$

Thereby, high-frequency components or abrupt block maxima dissipate rapidly compared to low-frequency components or smooth changes. Our findings shed light on a fundamental limitation of diffusion models while modeling block maxima and underscore the need for tailored approaches to preserve the block maxima consistent with the other values and to address the accurate representation of block maxima distributions.

We now provide the proof for Theorem 2 in the main paper.

**Proof for Theorem 2**

*Proof.* Using Lemma 1 and Assumptions 1, 2, and 3, we can prove the theorem as follows: For low frequencies, i.e., $\omega_k \to 0$, taking the limit as $\omega_k \to 0$ on the expression for the signal power spectral density $\overline{\mathcal{S}_n^f}(\omega)$, we have:

$$\lim_{k \to 0} \overline{\mathcal{S}_n^f}(\omega_k) = \lim_{k \to 0} \sigma_n^2 \cdot \exp(-\alpha_n |\omega_k|^{\beta_n}) = \sigma_n^2$$

since $\exp(-\alpha_n |0|^{\beta_n}) = \exp(0) = 1$. Therefore, as $k \to 0$, $\overline{\mathcal{S}_n^f}(\omega_k) = |f_n^k|^2$ approaches the variance $\sigma_n^2$. So, we can write:

$$\lim_{k \to 0} \overline{\mathcal{S}_n^f}(\omega_k) = \lim_{k \to 0} |\overline{f_n^k}|^2 = \sigma_n^2 = \zeta + \sigma_{\epsilon_n}^2 = \sigma_0^2 + \sum_{i=1}^{n-1} \sigma_{\epsilon_i}^2 + \sigma_{\epsilon_n}^2 = \sigma_0^2 + \sum_{i=1}^{n} \sigma_{\epsilon_i}^2$$

Similarly, for high frequencies, i.e., $k \to k_{\max}$, taking the limit as $\omega \to \omega_{\max}$ on the expression for the signal power spectral density $\overline{\mathcal{S}_n^f}(\omega_k)$ (Eq. (13)), we have:

$$\lim_{k \to k_{\max}} \overline{\mathcal{S}_n^f}(\omega_k) = \lim_{k \to k_{\max}} \sigma_n^2 \cdot \exp(-\alpha_n |\omega|^{\beta_n}) \to \sigma_n^2 \cdot \delta'$$

where $\delta' = \delta \cdot \gamma$

As $k \to k_{\max}$, $\overline{\mathcal{S}_n^f}(\omega_k)$ approaches $\sigma_n^2 \cdot \delta \cdot \gamma$. We can also write:

$$\lim_{k \to k_{\max}} |\overline{f_n^k}|^2 = \delta \cdot \gamma \cdot \left( \zeta + \sigma_{\epsilon_n}^2 \right) = \delta \cdot \gamma \cdot \left( \sigma_0^2 + \sum_{i=1}^{n} \sigma_{\epsilon_i}^2 \right)$$

Taking the ratio of high-frequency and low-frequency components after perturbations yields:

$$\frac{\lim_{k \to k_{\max}} |\overline{f_n^k}|^2}{\lim_{k \to 0} |\overline{f_n^k}|^2} = \delta \cdot \gamma$$

$\square$

**Proof for Theorem 3**

*Proof.* The proof begins by expressing the negative log-likelihood of the joint distribution $-\log p_\theta(\mathbf{x}_0, y_0)$ in terms of conditional probabilities:

$$
\begin{aligned}
-\log p_\theta(\mathbf{x}_0, y_0) &= -\log p_\theta(\mathbf{x}_0|y_0) \cdot p_\theta(y_0) = -\log p_\theta(\mathbf{x}_0|y_0) - \log p_\theta(y_0) \\
&\leq D_{\mathrm{KL}}(q(\mathbf{x}_{1:N}|\mathbf{x}_0, y_0)\|p_\theta(\mathbf{x}_{1:N}|\mathbf{x}_0, y_0)) \\
&\quad - \log p_\theta(\mathbf{x}_0|y_0) - \log p_\theta(y_0) \\
&= \mathbb{E}_q\left[ \log \frac{q(\mathbf{x}_{1:N}|\mathbf{x}_0, y_0)}{p_\theta(\mathbf{x}_{0:N})} + \log p_\theta(\mathbf{x}_0, y_0) \right] \\
&\quad - \log p_\theta(y_0) \\
&= \mathbb{E}_q\left[ \log \frac{q(\mathbf{x}_{1:N}|\mathbf{x}_0, y_0)}{p_\theta(\mathbf{x}_{0:N})} \right] - \log p_\theta(y_0)
\end{aligned}
\tag{19}
$$

$\square$

## D  Data

We performed our experiments using the following datasets. **(1) Synthetic Data (AR2)**: AR(2) dataset comprises synthetic time series data generated using an autoregressive model of order 2. **(2) Financial Data (Stocks)**: It features continuous-valued and aperiodic sequences, such as daily historical Google stocks data spanning from 2004 to 2019. We consider the adjusted closing price data for this work. **(3) Energy Data (Appliance Energy):** The UCI Appliances energy prediction dataset [3] encompasses multivariate, continuous-valued measurements. We consider appliance energy data for analysis. **(4) Weather/Climate Data (Daily Minimum Temperature)**: This dataset [17] comprises daily minimum temperatures in Melbourne, Australia, from 1981 to 1990. **(5) Medical Data (ECG5000: Congestive Heart Failure)**: The original dataset [9] for "ECG5000" originates from a 20-hour long electrocardiogram (ECG) obtained from the Physionet database. Specifically, it is derived from the BIDMC Congestive Heart Failure Database (chfdb), with the record labeled as "chf07." The processed data encompasses 5,000 heartbeats randomly selected from the original dataset.

## E  Experimental Results

Table 4 reports the evaluation of all values in time series using JS Divergence and predictive score, comparing the performance of our proposed method against baseline methods.

**Table 4** Comparison of generated samples' (all values) JS Divergence and Predictive Score using the baselines methods. **Bold** and Underlined entries denote the best and second-best result.

| Metrics | Methods | AR1 | Stock | Energy | Temperature | ECG |
|---|---|---|---|---|---|---|
| JS Divergence | beta-VAE | 0.0013±0.0004 | 0.0091±0.0011 | 0.0091±0.0008 | 0.0012±0.0002 | 0.0015±0.0003 |
| | c-beta-VAE | 0.0011±0.0004 | 0.0088±0.0012 | 0.0089±0.0007 | 0.0010±0.0001 | 0.0014±0.0002 |
| | TimeVAE | 0.0010±0.0003 | 0.0086±0.0008 | 0.0081±0.0011 | 0.0009±0.0001 | 0.0014±0.0001 |
| | TimeGAN | 0.0015±0.0004 | 0.0092±0.0015 | 0.0082±0.0009 | 0.0011±0.0003 | 0.0018±0.0003 |
| | cGAN | 0.0012±0.0002 | 0.0091±0.0010 | 0.0085±0.0008 | 0.0008±0.0001 | 0.0015±0.0002 |
| | RealNVP | 0.0014±0.0003 | 0.0093±0.0011 | 0.0092±0.0008 | 0.0010±0.0002 | 0.0017±0.0001 |
| | Fourier-Flows | 0.0013±0.0004 | 0.0087±0.0012 | 0.0083±0.0007 | 0.0009±0.0001 | 0.0015±0.0002 |
| | DDPM | **0.0007±0.0001** | 0.0061±0.0007 | 0.0058±0.0005 | **0.0005±0.0001** | 0.0010±0.0001 |
| | Diffusion-TS | 0.0008±0.0001 | **0.0057±0.0008** | 0.0061±0.0005 | 0.0008±0.0001 | **0.0009±0.0001** |
| | FIDE (Ours) | 0.0008±0.0001 | 0.0068±0.0010 | **0.0056±0.0006** | 0.0006±0.0002 | 0.0011±0.0001 |
| Predictive Score | beta-VAE | 0.8121±0.0410 | 1.0915±0.0215 | 0.8515±0.0104 | 0.8021±0.0109 | 0.9911±0.0133 |
| | c-beta-VAE | 0.7951±0.0555 | 1.0841±0.0121 | 0.8442±0.0110 | 0.7958±0.0089 | 0.9891±0.0151 |
| | TimeVAE | 0.7714±0.0345 | 1.0662±0.0211 | 0.8394±0.0089 | **0.7821±0.0105** | **0.9822±0.0101** |
| | TimeGAN | **0.7514±0.0451** | **1.0621±0.0198** | **0.8379±0.0151** | 0.7856±0.0098 | 0.9862±0.0125 |
| | cGAN | 0.7694±0.0354 | 1.0721±0.0188 | 0.8433±0.0181 | 0.7905±0.0122 | 0.9874±0.0151 |
| | RealNVP | 0.8011±0.0384 | 1.0914±0.0178 | 0.8533±0.0154 | 0.8033±0.0135 | 0.9981±0.0201 |
| | Fourier-Flows | 0.7985±0.0324 | 1.1008±0.0205 | 0.8501±0.0151 | 0.7988±0.0140 | 0.9954±0.0188 |
| | DDPM | 0.7711±0.0441 | 1.0751±0.0184 | 0.8488±0.0133 | 0.7912±0.0125 | 0.9925±0.0167 |
| | Diffusion-TS | 0.7684±0.0405 | 1.0722±0.0189 | 0.8501±0.0125 | 0.7889±0.0129 | 0.9910±0.0155 |
| | FIDE (Ours) | 0.7651±0.0488 | 1.0692±0.0192 | 0.8458±0.0151 | 0.7895±0.0135 | 0.9852±0.0158 |

