# OpenReview forum: "FIDE: Frequency-Inflated Conditional Diffusion Model for Extreme-Aware Time Series Generation"
_NeurIPS.cc/2024/Conference — NeurIPS 2024 poster_

### Official Review · Reviewer_8C1A · 2024-07-04

**Soundness:** 3
**Presentation:** 4
**Contribution:** 3
**Rating:** 7
**Confidence:** 4

**Summary:**

The problem statement, solution and its mathematical formulation is nicely presented in the draft.

Authors have identified a problem in the time series generation process which is under explored and they propose a new algorithm Frequency-Inflated Conditional Diffusion Model (FiDE) to address this. The problem lies in the modelling of block maxima which are related to extreme events in time series data where traditional diffusion models fail or does not take into account properly.

Authors demonstrated that these extreme values correspond to high frequency fourier components in time series and proposed the algorithm to inflate the weights of high frequency fourier components. The inflation of high-frequency components in the Fourier domain to prevent their premature dissipation during the diffusion process. This approach enables the model to retain significant extreme values that are typically lost in traditional modeling techniques.

The paper presents detailed performance comparison with several classes of models, e.g. VAE, diffusion, and GAN using datasets from several domains. The comparison is extended to 4 metrics explaining the motivation to use them.

**Strengths:**

- The problem statement and its novel solution are the strength of the research presented. It is of great importance to model the tails of time series distribution which are generally not very well modeled.
 - correctly identifying that the tail is mainly due to high frequency Fourier components and authors model it well using the algorithm proposed. Strength lie in the fact that they found the root cause and then proposed the solution.

**Weaknesses:**

The results [Fig 5] show the comparison of block maxima values for only one dataset. More details on the dataset and more comparison would be helpful to confirm the generality of the proposed algorithm.

Figure 5 shows the final results. It shows the improvement in the block maxima modelling but at the same time it does show degradation in modelling the bulk. An additional comparison of ALL - block maxima with a quantitative estimate of degradation in the performance is needed to understand the full impact of FIDE.

The table shows the comparison of metrics for block maxima because this is the main focus of this paper. Understood! But at the same time to make it a fair comparison the quantification on the ALL - maxima block is also needed to fully evaluate the model. With the present table and figures it is challenging to quantify if the model is doing "more good" than "bad".

Authors do mention in Results section L 257-261 that DDPM has slightly better performance but it is not quantified.

**Questions:**

The paper is generally well-written, but a few important points remain unanswered, which piques my curiosity. They are as follows:

- When defining the block maximum what is the time window? What criteria defined this time window? Is there a thumb rule or model to choose it dynamically?

- A discussion on error (unless missed) is needed. How are errors estimated on the metric reported in table 1. What are these errors (statistical/systematic or something else)?

**Limitations:**

- More experimental comparison like figure 5 are really needed in the paper draft for further clarity.
- What is the effect of FIDE on all samples (not just block maxima), a number would be useful to quote similar to block maxima.

---

> ### Author Rebuttal · Authors · 2024-08-07
>
> **Response to Question 1:**
>
> The choice of time window for defining block maxima is domain-dependent and application-specific, driven by the temporal characteristics of the data and the phenomena of interest. For climate or energy data, monthly block maxima (30-day or 90-day windows) are often relevant, capturing sub-seasonal to seasonal extremes. ECG data may require hourly or smaller windows to detect critical cardiac events, while financial data typically uses monthly or quarterly windows, aligning with reporting periods and long-term trends. The optimal window size should be determined based on the specific research questions, data resolution, and relevant time scales of the studied phenomena.
>
> **Response to Question 2:**
>
> We thank the reviewer for the question. The errors are estimated statistically. Each experiment was repeated five times, with the standard deviation across these iterations serving as our error estimate. This approach accounts for variability in model performance due to stochastic elements, such as random initialization of the network in training. We will explicitly detail this methodology in the revised manuscript to enhance clarity and reproducibility.
>
> **Response to Weaknesses:**
>
> Although our focus is on block maxima, our model generates the entire time series. We provided a comparison of the distributions for all values in Figure 5. However, due to space constraints, we were unable to include comparative figures for all baseline methods. Nevertheless, we have now included a table comparing the performance for all values (similar to *Table 1*). This table (*Table A2*) is provided in the attached PDF (please see the general rebuttal), which shows the metrics comparing the distribution and predictive performance for all values. The results indicate that our method is quite comparable to DDPM, though slightly worse, due to the tradeoff between preserving the overall vs extreme value distribution. However, it is still better than other VAE-based, GAN-based, and flow-based methods.

---

### Official Review · Reviewer_4b7m · 2024-07-07

**Soundness:** 2
**Presentation:** 2
**Contribution:** 2
**Rating:** 3
**Confidence:** 5

**Summary:**

This article discusses the shortcomings of the insufficient ability to focus on maximum values when applying DDPM in the field of time series generation, and proposes a new framework to overcome this problem by introducing a high-frequency expansion strategy in the frequency domain to ensure the emphasis on high-frequency values. At the same time, the article also proposes a generative modeling method based on conditional diffusion.

**Strengths:**

1. The logic of the article is relatively clear, the language is relatively clear and precise, and it is easy to understand.

2. The experimental settings of the article are relatively clear and the results are well organized.

**Weaknesses:**

1. The article does not set up an ablation experiment to explore the effects of the improved strategy.

2. The article has a small amount of experiments and incomplete research.

**Questions:**

1. In Chapter 3, the article mentioned that adding linear noise to the diffusion model will reduce high-frequency components faster. There are many ways to add noise in the diffusion model, such as the Noise Schedule in iDDPM. Is there any review on this part? Discussion and experimentation?

2. Should the Transformer in Figure 4 in Chapter 4 be a Transformer Encoder?

3. In the field of time series generation, the conditional generation of diffusion models has been studied in the article DiffusionTS. What is the difference between the two?

4. In the selection of comparative algorithms, there have been some time series generation methods based on flow methods, eg: Fourier Flow. Do articles based on this method have good performance in the distribution of block maximum values?

5. The article has mostly discussed the issue of block maximum value distribution in time series generation. Should we discuss the performance of the data generated by enhancing the block maximum value distribution in downstream tasks?

---

> ### Author Rebuttal · Authors · 2024-08-07
>
> **Response to Question 1:**
>
> We initially evaluated linear, sqrt, and sigmoid noise schedulers. We chose the linear scheduler as it provides a more gradual perturbation compared to the sqrt and sigmoid, which alter data more rapidly in initial iterations. We appreciate the reviewer's suggestion regarding iDDPM's noise schedule. Investigating the impact of alternative schedulers, such as the cosine scheduler in iDDPM, on high-frequency component dissipation is indeed an intriguing avenue for research. While beyond the scope of the current study, we acknowledge this as a valuable direction for future work and will explore it in subsequent investigations.
>
> **Response to Question 2:**
>
>  Yes, it is a Transformer Encoder.  We will change the notation/figure in the revised manuscript accordingly.
>
> **Response to Question 3:**
>
> Our proposed framework is designed to improve the generation of extreme values (i.e., block maxima) in time series whereas DiffusionTS enhances time series generation by emphasizing interpretability and the realistic representation of the generated data. Due to the tradeoff between preserving the overall distribution and accurately representing extreme values, DiffusionTS is insufficient to reproduce the extreme value distribution. To support this, we have included experiments comparing DiffusionTS to our method in the attached general rebuttal PDF (See *Table A1*). The results indicate that while DiffusionTS effectively captures the general distribution, it does not adequately preserve the extreme values.
>
> **Response to Question 4:**
>
>  We appreciate the reviewer's suggestion to include flow-based methods in our comparative analysis. In response, we have incorporated two such baselines: Fourier-Flows and RealNVP. The results of this extended comparison are presented in the attached *Table A1* (general rebuttal PDF). Our findings suggest that the proposed method consistently outperforms both flow-based baselines. Moreover, while flow-based methods perform comparably to the VAE- and GAN-based methods, they generally fall short when compared to diffusion-based approaches. This comparison further validates the efficacy of our proposed method across a broader spectrum of state-of-the-art approaches.
>
> **Response to Question 5:**
>
> The Predictive Score metric used in (Yoon et al., 2019) in *Table 1* directly addresses the performance of generated data in a predictive downstream task. This metric evaluates how well a predictive model trained on generated data performs when tested on real data, effectively assessing the fidelity of our generated time series in practical downstream applications. Our method achieves top performance on three datasets and second-best on two others. This underscores the practical utility of our approach beyond mere distribution matching, showing its value in generating data that preserves important predictive characteristics of the original time series.
>
> **Regarding Weakness 1:**
>
>  Indeed, we have provided an ablation experiment (See Appendix *Table 2*) exploring the effects of different strategies/modules of our proposed framework. This analysis specifically illustrates how different components of our approach contribute to the overall performance. These results provide valuable insights into the relative importance of each element in our model

---

> > ### Comment · Reviewer_4b7m · 2024-08-13
> >
> > Thanks. I am not satisfied with the author's rebuttal. there are still a lot of issues that need to be clarified here.
> >
> > Q1: The motivation for using this strategy requires further analysis and validation.
> >
> > Q3: Based on the numerical experimental results, it is difficult to determine whether the performance improvement of the proposed method compared to diffusionTS is due to capturing extreme values. The differences between the two methods need further explanation. Why can't diffusionTS capture extreme values, but your method can? What is the motivation and principle behind this? You should use Wilcoxon-Holm analysis the results in Table 1 to demonstrate the advantages and disadvantages of the proposed method.
> >
> > Q4: What were the experimental setups for diffusionTS, Fourier-Flows, and RealNVP? How is the fairness of the comparison ensured?
> >
> > W1: Wilcoxon-Holm analysis should be used to demonstrate that the results in Table 2 indeed show a significant performance improvement. The mean values alone do not sufficiently indicate that your results are superior. I also do not think it is a good idea to place the ablation experiments in the supplementary materials.
> >
> > The core challenge in time series generation tasks is the scarcity of data, yet diffusion models require large amounts of data for training. Please verify the performance of the proposed method on small-scale datasets, such as 10% of the stock dataset. I am certain that diffusionTS cannot handle this; please provide an explanation and validation. What is the significance of generating time series in the context of large-scale data?

---

> > > ### Author Response · Authors · 2024-08-14
> > >
> > > **Regarding Q3:**
> > >
> > > We thank the reviewer for suggesting the Wilcoxon-Holm test to further validate the performance improvement of our proposed method relative to DiffusionTS. Although we typically use t-test statistics, we have conducted the Wilcoxon-Holm test as recommended. This analysis was performed on results from three datasets across four performance metrics for both our method and DiffusionTS, which serves as the second-best baseline in most cases. Out of 12 comparisons (3 datasets × 4 metrics), our method demonstrated statistically significant performance improvements over DiffusionTS in 10 cases, with p-values less than 0.05. For the other two cases, the p-values were 0.09 and 0.154.
> > >
> > > DiffusionTS struggles to effectively capture extreme values due to its primary focus on generating and reconstructing entire time series, rather than specifically preserving the distribution of extreme values. According to the objective function (Eq. 10) of DiffusinTS, it aims to minimize mean squared error in the time domain and frequency domain. This approach leads the model to prioritize the central tendency of the data, resulting in a strong focus on predicting the conditional expectation. However, this focus tends to underrepresent extreme values, which are often located in the tail of the distribution. Our proposed method addresses this limitation by incorporating both mean squared error and a Generalized Extreme Value (GEV) loss (see Eq. 11 in our paper). This combination allows our model to simultaneously maintain overall accuracy and accurately capture the distribution of extreme values, which is critical for applications that rely on modeling rare events.
> > >
> > > Furthermore, DiffusionTS's objective function (Eq. 10) and methodology do not adequately account for the preservation of high-frequency components, which are crucially linked to extreme values, as demonstrated both empirically and theoretically in our paper. The mean squared error applied to Fourier coefficients in DiffusionTS treats low and high frequency components equally. However, time series are typically dominated by low-frequency components, with most of the energy or power spectral density concentrated in these regions. High-frequency components, despite their importance for extreme values, contain much less energy. Consequently, this equal treatment often results in overlooking or underemphasizing these critical high-frequency components. Our proposed method addresses this limitation by introducing a targeted strategy to inflate high-frequency components. This approach ensures that these components do not prematurely fade out during the diffusion model's noising process, allowing them to dissipate at a rate comparable to low frequency components, thus better preserving the characteristics of extreme values.
> > >
> > > **Regarding Q4:**
> > >
> > > We employed comparable experimental setups for DiffusionTS, Fourier-Flows, and RealNVP as we did for our proposed method and other baseline approaches. To ensure a fair comparison, we carefully tuned the general hyperparameters (number of epochs, learning rate, batch size) for all methods under evaluation. To account for variability and ensure fairness in our comparisons, we conducted 5 independent runs for each method. We then reported the mean and standard deviation of all performance metrics across these runs.
> > >
> > > **Regarding the last comment:**
> > >
> > > We appreciate the reviewer's comment about the challenges of data scarcity in time series generation tasks. However, it's important to clarify that our research addresses a distinct problem: preserving extreme values in time series generation. This preservation of extreme values remains crucial regardless of whether the underlying context involves small-scale or large-scale data. We acknowledge the reviewer's suggestion to verify our method's performance on small-scale datasets. While this is indeed an interesting direction for further investigation, given the limited time remaining for authors' response, we were unable to conduct such an analysis at this time. Finally, as shown in the paper (figure 5), while current diffusion models can effectively capture the general pattern of a time series in the context of large scale data, they fail to capture the distribution of extreme values. Effective generative modeling of the extreme values is significant for developing robust risk management strategies and enhancing disaster preparedness measures, as noted in the introduction.

---

### Official Review · Reviewer_Nq9n · 2024-07-11

**Soundness:** 3
**Presentation:** 3
**Contribution:** 3
**Rating:** 7
**Confidence:** 3

**Summary:**

The article introduces FIDE (Frequency Inflated Diffusion Estimation), which is geared towards better capturing
extreme values when generating time series through diffusion models, which the authors stress as crucial in domains like climate
science and disaster preparedness.

The approach involves inflating high-frequency components and conditionally generating samples based on block maxima, integrating
the GEV distribution to ensure fidelity in extreme event representation.

The authors run experiments comparing their approach to GANs and VAEs across a diverse set of datasets (AR1, Stock, Energy, Temperature, ECG).
They compare both the overall data distribution and extreme values and find their approach to show promising performance.

**Strengths:**

The problem statement and proposed solution of the paper are clearly laid out and mathematically described. The authors also propose a meaningful experiment setup to empirically test their approach against sensible baselines.

**Weaknesses:**

It is not clear to me from the presented results whether there is a trade-off between being able to capture the overall distribution of values well and capturing extreme values of the distribution. The experiments seem to suggests that that might be the case, but it could partially just be a matter of picking a different loss-function during selection of the hyperparameter for the GEV loss. It would be good to get more clarity on this aspect.

**Questions:**

What would be a model or situation where Assumption 1 is violated? I.e. the statement is that block maxima are related to high frequency components in _many_ real-world time series. Are there any notable exceptions and would one need a completely different modeling approach to capture those, or would your method still allow one to sensibly generate those time series even though the Assumption is violated?

**Limitations:**

The paper adequately addresses its limitations and does not appear to have negative societal impact.

---

> ### Author Rebuttal · Authors · 2024-08-07
>
> **Response to Questions:**
>
> We thank the reviewer for the question. While Assumption 1 holds for numerous real-world time series, it may not be applicable to slowly varying time series without abrupt changes, where block maxima build up smoothly.
> In such instances, a customized diffusion model like ours may not be necessary, as the block maxima should dissipate at a rate similar to other values during the noise addition process. To demonstrate this, we modified the example time series shown in Figure 3 of the paper by applying a moving average to smooth the data. We then applied the forward pass of diffusion model to the smoothed time series. As illustrated in the attached Figure 1 (see the attached PDF in the general rebuttal), both the block maxima and other (non-maxima) values of the smoothed time series dissipated at a similar rate, making Assumption 1 no longer valid. In this situation, the slow evolution of the block maxima is part of the general pattern of the time series. Thus, existing diffusion models should be sufficient to capture the block maxima without the need for our model.
>
> **Response to Weakness:**
>
> Indeed, our results reveal a trade-off between accurately capturing the overall distribution and effectively modeling the extreme values. This is evident in Figure 5 which shows that our proposed method excels in capturing the block maxima distribution while slightly underperforms in capturing the distribution of all values.
> To further validate this observation, we conducted an experiment using temperature data, evaluating the trade-off with different inflation weights (1.0, 1.15, 1.3) applied to high-frequency components. An inflation weight of 1.0 indicates no inflation, while 1.3 denotes inflating the high-frequency components by a factor of 1.3. We observed that decreasing the inflation weight from 1.3 to 1.15 and then to 1.0 generally resulted in decreased performance in capturing the block maxima distribution while improving the performance in capturing the overall distribution.
> Our ablation study (Table 2, Appendix) further supports this finding, showing that removing the GEV loss leads to lower performance compared to our proposed framework.  In short, these experiments demonstrate that the task of balancing the trade-off between capturing the overall distribution against the extreme values can be achieved by selecting the appropriate inflation weights (through cross-validation) and incorporating a GEV loss into the objective function.

---

### Official Review · Reviewer_ggGX · 2024-07-14

**Soundness:** 2
**Presentation:** 2
**Contribution:** 3
**Rating:** 5
**Confidence:** 4

**Summary:**

The paper presents a novel generative model designed to better capture extreme values in time series data. FIDE introduces a high-frequency inflation strategy to prevent the loss of extreme values, integrates conditional diffusion modeling to condition on block maxima, and incorporates the Generalized Extreme Value (GEV) distribution to ensure the accuracy of extreme value representation. Empirical results show that FIDE outperforms existing methods in maintaining the distribution of extreme events across various datasets, making it a practical application in the generative modeling of time series data.

**Strengths:**

1. Maintaining extreme value in generated time series data is an important but challenging problem. The proposed method can potentially satisfy this urgent need in real applications.

2. The problem is well motivated in Section 3, and the proposed method is reasonable and explained clearly.

3. The experimental results demonstrate that the proposed method can indeed preserve statistical information of extreme values.

**Weaknesses:**

1. Although inflating high-frequency signals is intuitive, the possible influence on the fidelity of generated time series is not discussed.

2. The compared baselines, especially diffusion-based approaches, are not enough. For example, there are several works developing diffusion models for general time series generation (Ref-1), or domain-specific time series generation (Ref-2,3). Also, I think those diffusion-based probabilistic time series forcasting approaches can also be used for generation.

Ref-1 Yuan, X., & Qiao, Y. (2024). Diffusion-ts: Interpretable diffusion for general time series generation. ICLR24

Ref-2 Kong, Z., Ping, W., Huang, J., Zhao, K., & Catanzaro, B. (2020). Diffwave: A versatile diffusion model for audio synthesis. arXiv preprint arXiv:2009.09761.

Ref-3 Zhou, Z., Ding, J., Liu, Y., Jin, D., & Li, Y. (2023, November). Towards generative modeling of urban flow through knowledge-enhanced denoising diffusion. Sigspatial23.

**Questions:**

Please answer my listed weaknesses above.

**Limitations:**

None.

---

> ### Author Rebuttal · Authors · 2024-08-07
>
> **Response to Question 1:**
>
> We appreciate the reviewer's concern regarding the fidelity of generated time series. We have investigated this issue in our experiments. First, Figure 5 demonstrates the tradeoff between preserving the overall distribution and the distribution of block maxima. The results suggest  that our model excels in capturing block maxima, with minimal alteration to the overall series. In contrast, the baseline DDPM approach, which is capable of replicating the overall distribution, significantly underestimates the distribution of block maxima values. The analysis and discussion provided in Lines 257-261 highlights this fidelity aspect. While we did not provide the full quantitative results (similar to *Table 1*) for the entire time series due to space constraints, we have included them in the attached *Table A2* (provided in the general rebuttal PDF) and will include them in the appendix of the revised paper.
>
> Finally, the predictive score metric shown in *Table 1* also shows the fidelity of the generated time series by utilizing them for downstream time series forecasting tasks. Following a similar approach as in (Yoon et al, 2019), the downstream prediction task here corresponds to a multi-step time series forecasting task, which includes both predicting the block maxima and non-maxima values in a forecast window.  The results suggest that our method achieves the best performance on three datasets and the second-best on two others, underscoring the model’s ability to accurately reproduce the temporal properties of the time series.
>
> **Response to Question 2:**
>
> We acknowledge the reviewer's valid point regarding the additional baseline methods. While our initial selection encompassed various generative model types, we concur that incorporating additional diffusion-based baselines is warranted, given our proposed method's foundation in diffusion models. In response to this insightful suggestion, we have expanded our comparative analysis to include the recent Diffusion-TS model, as recommended by the reviewer.
>
> The results of this extended comparison are summarized in attached *Table A1* (provided in the general rebuttal PDF). Our findings conclusively demonstrate that our proposed method continues to outperform all baseline methods, including these new additions. Notably, Diffusion-TS emerges as the second-best baseline in most scenarios, underscoring the efficacy of diffusion-based approaches in this domain.

---

### Author Rebuttal · Authors · 2024-08-07

Thank you all for your careful and valuable suggestions. In response to your insightful feedback, we have expanded our comparative analysis to include the latest Diffusion-TS model and two additional flow-based baselines (Fourier-Flows and RealNVP) as recommended by the reviewers. This ensures a more comprehensive evaluation. We have also addressed your other suggestions and clarified the points you raised in the individual rebuttal.

Please find the attached PDF that contains 1 figure and 2 tables to address reviewers' comments.

---

### Decision · Program_Chairs · 2024-09-25

**Decision:**

Accept (poster)

**Comment:**

This paper introduces FIDE (Frequency Inflated Diffusion Estimation), a novel approach for generating time series data that better captures extreme values. The method combines high-frequency inflation, conditional diffusion modeling, and integration of the Generalized Extreme Value (GEV) distribution.

Strengths highlighted by the reviewers:
- Addresses a significant and underexplored problem in time series generation.
- Innovative combination of techniques: high-frequency inflation, conditional diffusion modeling, and GEV distribution integration.
- Comprehensive experiments across various datasets demonstrating improved performance.
- Clear presentation and mathematical formulation.

Weaknesses:
- Some questions about comparisons with recent state-of-the-art diffusion models.
- Potential trade-off between capturing overall distribution and extreme values could be further explored.

The paper presents a novel and important contribution to the field of time series generation, with a focus on preserving extreme values. The majority of reviewers recognize its significance and technical soundness. While some concerns were raised, the authors have adequately addressed most of them in their rebuttal.